# When Do Individuals Get More Injured? Relationship between Physical Activity Intensity, Duration, Participation Mode, and Injury

**DOI:** 10.3390/ijerph182010855

**Published:** 2021-10-15

**Authors:** Ju-Pil Choe, Ji-Su Kim, Jeong-Hui Park, Eunhye Yoo, Jung-Min Lee

**Affiliations:** 1Global Campus, Graduate School of Physical Education, Kyung Hee University, Yongin-si 17014, Korea; jupilchoe@khu.ac.kr (J.-P.C.); jisukim98@khu.ac.kr (J.-S.K.); jeonghee@khu.ac.kr (J.-H.P.); 2Department of Physical Education, Seoul National University, Seoul 08826, Korea; yeh04@snu.ac.kr; 3Sports Science Research Center, Global Campus, Kyung Hee University, Yongin-si 17014, Korea

**Keywords:** physical activity, injury, participation mode, intensity, duration

## Abstract

The present study examines the association between physical activity (PA) participation modes (i.e., family, friends, club members, and alone), PA volumes (i.e., intensity and duration), gender, and injury. A total of 9000 Koreans aged 10–89 years participated in the Korean Survey of Citizens’ Sports Participation project. However, participants who did not respond to a question regarding PA participation modes (*n* = 2429) and those under 18 years old (*n* = 489) were excluded from this study. Analysis of variance (ANOVA) was used to compare the groups’ characteristics and the association between PA participation modes and injury was demonstrated by conducting multinomial logistic regression analysis. The risk of injury was significantly higher in the friend and club member groups than in the alone group. In addition, PA intensity and gender were critical risk factors of injury, while PA duration showed no significant results. The results indicated a strong association between PA participation modes, PA intensity, gender, and injury, and an interesting finding is that more injuries derived from a higher intensity of PA, not from a longer duration of PA. Therefore, this present study directly documented that considerable attention should be placed on the factors that affect injuries, such as gender and PA intensity, to prevent unpredictable injury and encourage potential participants to exercise with diverse participation modes and appropriate intensity.

## 1. Introduction

Participating in regular physical activity (PA) has been acknowledged as a vital factor in health-related outcomes [1,2]. It has been reported by various governments and institutions that regular PA of individuals has positive health consequences, and PA guidelines have emerged to encourage individuals to participate in PA [3,4,5,6]. However, according to the World Health Organization (WHO), 23% of adults worldwide still do not meet adequate PA levels recommended by the WHO (150–300 min of moderate-intensity PA or 75–150 min of vigorous-intensity PA, or some equivalent combination of moderate- and vigorous-intensity aerobic PA per week) [7]. Herazo-Beltrán et al. [8] demonstrated that one of the reasons why potential participants did not engage in PA was concern over the possibility of injuries. According to the study, 46.5% of respondents who participated in regular PA had a fear of injury or re-injury as a barrier to PA participation. Furthermore, various studies have shown that the main risk of PA, including aerobic and muscular training, is musculoskeletal injury, especially in individuals being inappropriately trained [9,10].

There is no doubt that PA has positive effects on individual health, but it also has negative effects, such as severe injuries (i.e., strain, sprain, fracture, or contusion). Any injuries that occur during PA are a deleterious consequence and happen in unpredictable circumstances [11,12]. For these reasons, various researchers have strived to help clinicians and trainers understand injury etiology and to develop preventive measures [13]. According to a prior study regarding the perception of injury in PA participants, Snodgrass et al. [14] found that a majority of participants thought they were physically active (81%) and confident (84%), but only 54% of the participants were aware of some strategies to prevent injuries. The study demonstrated that most individuals were barely aware of the risk of injury while participating in PA, which makes them get injured easily as an unintended consequence.

During PA, participation modes (i.e., family, friends, club members, and alone) are important factors to influence PA volumes (i.e., intensity and duration), and PA volumes are directly related to inducing injuries [15,16,17]. PA participation mode can be explained as a supporter, instructor, and partner. The different PA modes could affect individuals in a variety of ways: (1) instrumental/direct, (2) psychological/emotional, and (3) instructional/informative [18,19,20]. A study investigated the differences in PA partners by race, gender, and dimensions of PA (i.e., work, sports, and leisure) among young adults in the United States. The study found that Caucasian women’s PA volumes were positively related to friends but African American women’s PA volumes were more closely related to family members [21]. In addition, Gellert et al. [22] compared three groups who participated in more physical activity: (1) individuals whose partner also participates in PA, (2) individuals whose partner does not participate in PA, and (3) individuals participating in PA without a partner, and revealed that PA volumes were significantly higher in individuals whose partners also participate in PA. Although partners may not be a contributing factor in determining injuries, PA volumes could be greatly influenced by different PA partners, which are also likely to be associated with injuries.

Several studies have indicated that gender is also a critical factor when individuals participated in PA. One of the studies regarding participants’ motivation and context preferences showed significant differences in terms of gender, such as reasons for participating in PA, preferred environments of PA, and characteristics of PA [23]. In addition, Christopher et al. [24] explained that females’ anxiety sensitivity decreases their PA and substantially affects the gender differences in PA. In the same context, many studies found that males are more likely to suffer injuries during PA than females because they are more active [25,26]. Therefore, interventions reducing anxiety, sensitivity, and injury should be taken into account for the gender gap in PA participation.

Additionally, PA volumes are the main cause of inducing injuries. For instance, a previous study, examining walkers’ and runners’ injury rates, found that moderate-intensity and a shorter duration of PA have a lower risk of injury than vigorous-intensity and a longer duration of PA, but that low-intensity with a longer duration of PA is not significantly associated with an increased risk of activity-related injury [27]. Similarly, Malisoux’s study demonstrated that intensity during the week before an injury was significantly stronger compared to that of the four preceding weeks [28]. Those studies provide PA participants with recommendations to avoid excessive intensity or duration of PA to prevent injuries; however, limited studies have systematically investigated the association between PA participation modes and the rate of injuries.

Therefore, the primary aim of the current study is (1) to understand the differences in injury rates in each PA participation mode and (2) to examine the important association of whether the volumes (i.e., intensity and duration) and modes of physical activity affect injury occurrence.

## 2. Materials and Methods

### 2.1. Design

The Korean Survey of Citizens’ Sports Participation is a series of cross-sectional population studies conducted by the Ministry of Culture, Sports, and Tourism [29]. The PA participation modes, intensity and duration of PA, and the number of injuries were assessed using data from the Korean Survey of Citizens’ Sports Participation in 2019. The questionnaire has been examined for validity and reliability by experts in the National Statistical Office (NSO) [29].

### 2.2. Study Participants

This study utilized the Korean Survey of Citizens’ Sports Participation, which was conducted from September to November 2019 with 9000 Koreans above 10 years old. Stratified multistage cluster sampling using 17 regions, sex, and age groups was used in the survey to obtain a nationally representative sample from the whole Korean population (i.e., 50 million).

Among the participants, this study excluded some respondents who did not respond to the question regarding PA participation modes (*n* = 2429), and those under 18 years old were excluded (*n* = 489) to focus on the association in adults. Therefore, 6082 participants’ data were utilized for the analysis. Additionally, all of the participants were categorized into four groups, <40 (*n* = 2250), 41–50 (*n* = 1121), 51–70 (*n* = 2071), and >71 (*n* = 640). As the survey did not collect any private information from respondents, such as participants’ names, social security numbers, or home addresses, ethical approval was not required. All study procedures were approved by the Korean Ministry of Culture, Sports, and Tourism.

### 2.3. Variables

#### 2.3.1. Participation Modes in Physical Activity (PA)

All 6082 respondents answered a question (i.e., ‘Who do you mainly participate in PA with?’) in the study’s questionnaire. The question asked for respondents’ PA participation modes. The options of the answer were consisted of ‘family’ (*n* = 992), ‘friends’ (*n* = 1770), ‘sports club members’ (*n* = 348), ‘colleagues at work’ (*n* = 201), ‘local residents’ (*n* = 226), and ‘alone’ (*n* = 2545). In the present study, ‘colleagues at work’ (*n* = 201) and ‘local residents’ (*n* = 226) were merged into ‘friends’ (*n* = 2197) because those two options share similar characteristics with a friend, so could be considered as friends. Respondents that checked ‘family’ included participants who exercise with their parents, siblings, and children, and respondents who participate in PA with friends or acquaintances from their school, neighborhood, and working places belonged to ‘friends’. ‘Sports club members’ consisted of participants who enrolled in sports clubs or any small groups that have exercise as their specific purpose. Lastly, ‘alone’ means someone who exercises by oneself, without any partners.

#### 2.3.2. Injuries

All of the respondents also checked on an injury-related question (i.e., ‘Have you ever been treated at a hospital because you got injured while exercising in the latest year?’). It was answered by ‘yes’, which means ‘I have been treated at a hospital for an injury from PA more than once in the latest year’, or ‘no’, which means ‘I have never been treated in hospital for an injury from PA in the latest year’. Various types of injuries could be derived from PA participation, such as a sprain, fracture, or contusion, and the number of days for treatments has infinite variety depending on the type of injury. However, the present study has focused just on the occurrence of injuries, setting aside the severity of injuries and the number of days for treatment.

#### 2.3.3. Physical Activity (PA) Volumes

PA volumes included PA intensity and duration as an indicator to compare each PA participation mode. The study’s participants were asked to respond to PA-related questions (i.e., types, frequency, intensity, and average duration). In particular, there were the first, second, and third priorities in the sports category, and only the first major priority of sports was utilized. ‘Average duration’ was calculated in minutes, and ‘intensity of participation’ was checked among three options (i.e., light-, moderate-, and vigorous-intensity). PA was calculated by ‘metabolic equivalent task (MET) level × minutes × number of days per week’ for each intensity, with 2.0 METs for light intensity, 3.0 METs for moderate physical activities, and 6.0 METs for vigorous physical activities [30].

#### 2.3.4. Covariates

The present study was analyzed with several covariates, such as sex, age, education, marital status, family number, number of children, monthly income, and occupation. Age was divided into four groups (i.e., under 40, 41–50, 51–70, above 71) and education was categorized into five groups (i.e., no education, elementary school, middle school, high school, and above undergraduate and/or graduate). Marital status consisted of four groups (i.e., married, single, bereaved, and divorced). Household size was chosen from 1 to 5 and more than 6, and the number of children was from 1 to 3 and more than 4. Monthly income was classified into four quartile groups (Q1: ≤2,900,000, Q2: ≤4,000,000, Q3: ≤5,000,000, and Q4: >5,000,000 KRW). Occupation status was chosen between ‘I have’ or ‘I do not have’.

### 2.4. Data Analysis

The demographic information of the participants was summarized by descriptive statistics in SPSS 26.0 version (SPSS Inc., Chicago, IL, USA). Descriptive statistics were conducted to compare participants’ personal information (i.e., gender, age, education, marital status, household size, number of children, monthly income, and occupation) for the groups of participants (i.e., family, friends, club members, and alone). Analysis of variance (ANOVA) was also used to compare the groups’ characteristics with Bonferroni analysis. In addition, multinomial logistic regression was utilized to examine the association between injury occurrence and gender, PA participation modes, and PA volumes (i.e., intensity and duration), and compared injury occurrence in each variable’s subgroup. Results of this analysis are expressed as odds ratios (OR) with 95% confidence intervals (95% CI) and a statistical significance set by *p* < 0.05.

## 3. Results

Table 1 presents the participants’ demographic characteristics (i.e., gender, age, monthly income, education, marital status, household size, number of children, and occupation) using descriptive statistics (*n* = 6082), and the results are summarized in number and proportion in Table 1. Overall, the number of males and females showed non-significant number differences in family, friend, and alone groups, but the number of males was almost four times higher than that of females in the club member group. The number of participants in the under 50 years old group was relatively higher in all groups than that of the over 51 years old group, but there were no significant differences in each participation mode (*p* > 0.05). In regard to monthly income, more than half of each group earned more than 2900 thousand KRW (USD 2569), and the club member group revealed significant differences between each of the other groups (i.e., family, friend, and alone) (*p* < 0.001). There were no significant differences between the three groups (i.e., family, friend, and alone) (*p* > 0.05). Educational status indicated that participants up to college undergraduate and/or graduate have the most proportion in all groups (family: 45.76%, friend: 50.46%, club member: 42.52%, and alone: 46.27%). Furthermore, almost 70% of each groups’ participants were married and more than half of the participants had at least one child. There was an almost two-fold difference between occupational status in the alone, friend, and family group, and four-fold in the club member group.

Table 2 shows the demographic characteristics of the participants who had experienced an injury in a year. The number of males in the friend and club member group was three and seventeen times higher than females, respectively. In participants’ age, an age difference was found between the family group and two groups (club member and friend group). The result indicated that the age of the family group (mean = 59.00 years) was significantly higher than the club member (mean = 43.32 years) (*p* < 0.05) and the friend group (mean = 43.85 years) (*p* < 0.05), but there were no significant differences between the other pairs (*p* > 0.05). All groups’ monthly incomes also presented differences in Table 2. Half of the monthly income of the family and friend group was under 4 million KRW, and that of the club member and alone group was above 4 million KRW. The mean difference between the club member and two groups (friend and family) was 940 thousand KRW (*p* < 0.05) and 1800 thousand KRW (*p* < 0.01), respectively, and there were no significant differences between the other pairs. Additionally, although education, marital status, household size, and the number of children maintained the same pattern, the family and alone groups’ occupational status indicated inverse results as compared to the PA mode participants.

Table 3 shows the association of PA participation modes, PA volumes, and injury by using multinomial logistic regression analysis. The rate of injury occurrence in adults was significantly associated with gender. Specifically, the rate of injuries for males was 1.7 times higher than that for females (OR = 1.700; CI = 1.125–2.568; and *p* = 0.012). In addition, based on the alone group, the club member group’s participants were 5.2 times more likely to experience injuries (OR = 5.228; CI = 3.069–8.905; and *p* < 0.001), and participants in the friend group were also observed to be strong risk factors for predicting injuries (OR = 1.785; CI = 1.132–2.814; and *p* = 0.013). However, the family group’s participants showed no significant difference against the reference group (OR = 1.006; CI = 0.501–2.016; and *p* = 0.986). In addition, vigorous-intensity PA had a significant impact on injury occurrence as compared to light-intensity PA (OR = 4.594; CI = 2.483–5.501; and *p* < 0.001), but moderate-intensity PA presented no statistical difference (OR = 1.718; CI = 0.983–3.004; and *p* = 0.057). Gender, participation modes, and PA intensity also presented a strong association with injury occurrence; however, PA duration was not a significant risk factor associated with injuries (above 60 min: OR = 1.239; CI = 0.705–2.179; and *p* = 0.455–60 min: OR = 0.908; CI = 0.510–1.614; and *p* = 0.743).

Specifically, Figure 1 revealed the comparison of the number of injury occurrences (Figure 1A) and PA volume (Figure 1B) for each PA participation mode. According to Figure 1A, when males participated in PA with friends, there was the highest injury occurrence (*n* = 39), followed by the club members (*n* = 35), alone (*n* = 15), and family groups (*n* = 6). Meanwhile, the percentage of injury occurrence was highest in the club member group (12.7%), followed by the friend (3.4%), family (1.4%), and alone group (1.2%). In females, the club member group had the lowest number (*n* = 2), following the family (*n* = 5), friend (*n* = 13), and alone groups (*n* = 16). However, in a percentage context, the club member group (2.7%) had the highest percentage of injury occurrence followed by the friend and alone (1.2%) as well as family groups (0.9%). Figure 1B shows the comparison to the PA volumes (i.e., intensity and duration) by calculating the average of MET min/week in each participation mode group. The highest average METs were found in the club member group (males: 1062.84 ± 1052.22, females: 1193.84 ± 1212.17 MET min/week), and the lowest was in the family group (males: 578.53 ± 723.20, females: 424.65 ± 451.80 MET min/week), regardless of gender. Additionally, the alone group’s average METs (males: 853.25 ± 851.71, females: 667.09 ± 625.47 MET min/week) were followed by the friend group’s (males: 678.44 ± 896.99, females: 1193.84 ± 1212.17 MET min/week). By using Bonferroni post hoc analysis, significant mean differences (*p* < 0.01) were found between the club member and friend groups (males: 384.39, females: 527.74 MET min/week), club member and family groups (males: 484.30, females: 769.18 MET min/week), club member and alone groups (males: 209.58, females: 526.74 MET min/week), family and alone groups (males: 274.71, females: 242.43 MET min/week), friend and alone groups (males: 174.81 MET min/week), and friend and family groups (females: 241.43 MET min/week). Significant differences (*p* > 0.05) were not between the friend and family groups in males (mean differences = 99.90 MET min/week) and between the friend and alone groups in females (mean differences = 0.99 MET min/week).

## 4. Discussion

The purpose of this study was to examine the differences in injury rates in each PA participation mode and the critical association of whether the volumes (i.e., intensity and duration) and modes of PA affect injury occurrence.

The present study demonstrated that gender had a significant association with injury occurrence (*p* = 0.012, OR = 1.700) in particular, as males who participate in regular PA have a higher injury risk rate than females. Although males and females showed a similar prevalence of meeting the recommended guidelines, males spent more time on PA than females per week [31]. Not only the duration of PA but also males’ PA intensity is much higher than that of females [32,33]. According to a study regarding injuries of runners through the perspective of gender, males were at a higher risk than females for suffering from running-related injuries, particularly (1) in the case of having running experience of 0–2 years, (2) if they had a history of previous injuries, restarting running, and (3) if they had an excessive running distance of more than 40 miles per week [25]. These results might be from males being more active than females as well as the fact that they tend to participate in more competitive sports [34]. Additionally, the higher PA intensity and duration that males engage in may be potential risk factors of an increased injury risk rate. Thus, practitioners and supporters in fields should recognize gendered differences, such as the strengths, weaknesses, and capability to participate in PA, and compose appropriate PA plans in regard to gender in order to minimize the odds of injuries occurring. These changes will allow practitioners to run a program efficiently, and participants to engage in PA safely.

The most outstanding result in the current study was to differentiate the injury occurrence between the groups (i.e., family, friend, and club member groups) compared to the reference group (i.e., the alone group). Participants in the club member group had a five-fold higher injury occurrence rate and the friend group had a two-fold higher injury occurrence rate than participants who work out alone. On the other hand, the probability of the family group getting injured was the same as the alone group (family: *p* = 0.986, OR = 1.006). These outcomes are consistent with Weicong’s findings, and the study demonstrated that participants engaging in PA with sports team members were almost two times more vulnerable to getting injured than their counterparts [35]. Similarly, Gao’s study found that university and other sports team members were at a two- and three-times-higher risk of PA-related injury, respectively [36]. Furthermore, several previous studies have reported a positive association of participating in PA with friends and PA volumes (i.e., intensity and duration) in young and old adults [37,38]; PA volumes and PA-related injury also have a significant relevance [36,39]. Based on the results of Table 3 and Figure 1B, we speculated that participating in PA with sports club members and/or friends causes volumes of PA to increase, especially intensity, leading to the overburdening of muscles, bones, and nerves due to frequent contact with others via excessive movements [40,41]. Therefore, PA participants who exercise with other club members and/or friends should acknowledge the different risks of injury and prevent it in advance. Additionally, instructors, supporters, and leaders of sports clubs should educate participants on how they can avoid injuries through educational programs prior to PA. Furthermore, the association between PA participation mode and injury requires a further study on whether other PA participation that did not refer to the present study could potentially prevent or cause injuries.

Additionally, the PA intensity groups (light-, moderate-, and vigorous-intensity PA) showed that the vigorous-intensity PA group had an almost five-fold injury occurrence rate compared to the light-intensity PA group. In contrast, the moderate-intensity PA group had no significant difference from the light-intensity PA group. However, compared to the different PA duration groups (above 60 min, 60 min, and under 60 min), there were no crucial signs for injury occurrence. Although a few studies are insisting that PA duration is related to the increased rate of injury [42,43,44], those studies’ participants were differentiated from the present study because those studies’ participants were in various environments and had different occupations to the participants in the present study (i.e., military, athletes, and specific age ranges). The present study investigated the association between PA duration and injury in general participants, and the results could be interpreted to arrive at the conclusion that the participants should keep their PA intensity low so that they can maintain a longer PA duration without getting injured. Hootman et al. [27] demonstrated that the risk of injury did not increase with an increased duration of PA per week among walkers and sport participants. Conversely, among runners, the risk of activity-related injury increased with a longer duration of PA per week. By comparing the intensity (of walking and running), their study supports the results of the present study that PA intensity is a more critical risk factor for injury than PA duration. Therefore, participants engaging in PA should be aware of the differences in intensity and duration and pay more attention to the environment in order to prevent detrimental consequences. Additionally, if instructors, supporters, and leaders exist, they should adjust the intensity and duration of PA depending on the participants’ purpose and status.

The average MET min/week of both males and females presented a similar pattern (with club member > alone > friend > family). The club member group indicated the highest average MET min/week and was followed by the alone, friend group, and family groups. One of the reasons for the gender differences in the club member and friend groups can be explained by the sheer number difference of participation in sports clubs [45]. According to a study regarding motivation for adult participation in PA, males’ prior motivations were mastery of skills and enjoyment; on the other hand, females’ priorities were their psychological and physical condition in addition to their appearance [46]. In other words, males obtain pleasure by developing skills and exerting their skills in competitions, which is possible in sports clubs or with friends, while females take advantage of PA to satisfy themselves, which is possible without a PA partner. Many studies have insisted that the promotion of PA using sports clubs is paying off, so the promotion should be maintained and continually developed [47,48]. In a study by Ooms et al. [49], in particular, insufficiently active participants showed significantly increased PA volumes from baseline to six months later because they exercised as part of a sports club for that period. Therefore, in these circumstances, more specific recommendations and policies (e.g., educational and promotional programs) [50] for preventing males in sports clubs from suffering injuries (e.g., designating instructors or leaders in sports clubs) and for increasing female participation in sports clubs (e.g., opening female sports clubs and events) are needed to improve the PA environment for both genders.

Through the efforts of governments and institutions in many countries, it is well known that PA helps individuals who exercise regularly improve their health, so the majority of participants engage in PA for health benefits (i.e., a longer life, less illness, and well-being). Practically, one of the studies regarding the association between PA and cost reported that a $1 investment for exercise resulted in $2.94 in direct medical benefit. Because of these positive results of PA, participants might exercise more vigorously and frequently to obtain excessive benefits. However, as previously mentioned, increased volumes of PA are likely to increase the possibility of injury. A study concerning the association between PA and the cost from injury treatments showed that the average cost of treatment for a sports injury was $1510 per injury, which could offset the benefits of PA. Based on these studies, individuals should take into account the potential results derived from PA either positively or negatively, so that they can take only the benefits from participating in PA. The results of this study’s findings demonstrate that individuals can consider how they can minimize the cost of injury and how they can maximize the benefits of PA.

The present study had several positive strengths. First, none of the previous studies has investigated an association between injury and PA participation modes, in particular, within the family, friends, club members, and alone groups. In addition, PA intensity and duration were utilized to validate the reasons for the association between injury and PA participation modes. Furthermore, we re-emphasized the gender differences in PA by dividing the genders as part of calculating METs and the number of injury occurrences. Second, in contrast to previous studies whose participants were professional athletes, military individuals, or specific age ranges, the participants of the present study were general adult participants. In sum, these results can allow public participants who exercise in diverse environments to protect themselves from injury. However, there are some limitations in the current study. This study focused on injury occurrence, not the type or severity of injuries. We suggest that additional studies would examine the details of injuries and analyze the differences of the injuries in each participation mode. Likewise, types of PA were not considered in this study so that future researchers could investigate differences in injuries in various types of PA. In subsequent studies, more detailed information on the prevention of injury may be organized by studying differences in injuries in each type of PA. Additionally, self-report questionnaires were utilized to measure PA volumes instead of objective measures such as an accelerometer, leading to an under- or over-estimation of PA volumes. Additionally, participants of the current study were only Korean adults. Therefore, future studies need to investigate various ethnicities and appropriate sample sizes.

## 5. Conclusions

The present study investigated an association between PA participation modes and PA-related injury occurrence and found that there is a significant difference in injury occurrence in different participation modes. Interestingly, the PA intensity has a strong association with injury occurrence compared to the PA duration. Therefore, this present study directly documented that considerable attention should be placed on the factors that affect injuries, such as gender, PA intensity, and duration, in order to prevent unpredictable injury.

## Figures and Tables

**Figure 1 ijerph-18-10855-f001:**
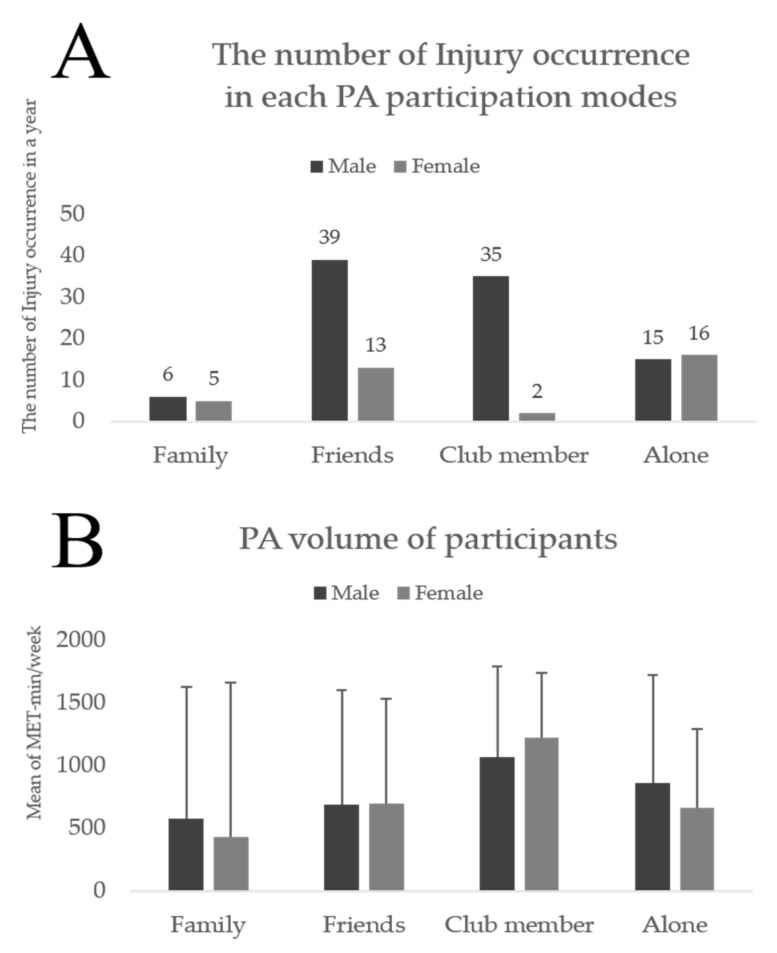
The comparisons of the number of injury occurrences in each PA participation mode (**A**) and the PA volumes of participants (**B**) among club member, friend, family, and alone groups. Error bars represent the 95% confidence interval.

**Table 1 ijerph-18-10855-t001:** Characteristics for participants in four groups.

Variables		Family (*n* = 992) ^A^		Friend (*n* = 2197) ^B^		Club Member (*n* = 348) ^C^		Alone (*n* = 2545) ^D^		*F*-Value	Post Hoc
		No. (%)	Mean ± SD	No. (%)	Mean ± SD	No. (%)	Mean ± SD	No. (%)	Mean ± SD		
Gender	Male	430 (43.34)		1153 (52.48)		275 (79.02)		1251 (49.15)			
	Female	562 (56.65)		1044 (47.51)		73 (20.97)		1294 (50.84)			
Age	≤40	359 (36.23)	28.71 ± 6.89	824 (37.53)	28.53 ± 6.72	106 (30.54)	30.24 ± 5.58	961 (37.81)	29.86 ± 6.07	1.15	N/A
	41–50	208 (21.03)	45.67 ± 2.59	372 (16.93)	45.45 ± 2.70	90 (25.92)	45.37 ± 2.47	451 (17.74)	45.53 ± 2.73		
	51–70	339 (34.23)	59.22 ± 5.67	743 (33.82)	59.35 ± 5.35	142 (40.81)	57.59 ± 4.70	847 (33.33)	59.84 ± 5.48		
	≥71	86 (8.61)	73.92 ± 3.40	258 (11.62)	74.82 ± 3.87	10 (2.83)	74.10 ± 2.28	286 (11.12)	74.73 ± 3.78		
Monthly income	<290	223 (22.47)	191.52 ± 58.23	540 (24.57)	187.89 ± 65.11	41 (11.78)	216.46 ± 57.75	658 (25.85)	187.97 ± 63.25	22.46 ***	C > A, B, and D
	290–400	346 (34.87)	350.30 ± 33.51	684 (31.13)	360.11 ± 35.76	98 (28.16)	352.80 ± 37.05	739 (29.03)	352.12 ± 36.99		
	400–500	288 (29.03)	459.25 ± 28.73	568 (25.85)	465.43 ± 30.11	93 (26.72)	471.50 ± 29.11	676 (26.56)	469.61 ± 29.33		
	>500	135 (13.60)	623.48 ± 118.97	405 (18.43)	621.12 ± 89.20	116 (33.33)	623.53 ± 83.31	472 (18.54)	612.53 ± 87.12		
Education	None	7 (0.70)		19 (0.86)		2 (0.57)		15 (0.58)			
	≥Elementary	51 (5.14)		126 (5.73)		12 (3.44)		130 (5.10)			
	≥Middle	69 (6.95)		162 (7.37)		116 (33.33)		193 (7.58)			
	≥High	411 (41.43)		781 (35.54)		70 (20.11)		927 (36.42)			
	≥College	454 (45.76)		1109 (50.46)		148 (42.52)		1280 (46.27)			
Marital status	Married	749 (75.51)		1491 (67.87)		257 (73.86)		1739 (68.33)			
	Not married	217 (21.88)		619 (28.17)		79 (22.71)		688 (27.04)			
	Others	26 (2.61)		77 (3.96)		12 (3.44)		115 (4.52)			
Household size	Single	28 (2.82)		186 (8.47)		26 (7.48)		162 (6.36)			
	Two	215 (21.68)		459 (20.89)		47 (13.51)		680 (26.72)			
	Three	225 (22.68)		530 (24.13)		88 (25.28)		630 (24.76)			
	Four	493 (49.69)		952 (43.33)		170 (48.85)		999 (39.26)			
	Five	29 (2.93)		64 (2.91)		17 (4.88)		64 (2.51)			
	More than six	2 (0.20)		6 (0.27)		0 (0.00)		10 (0.39)			
Number of children	No child	272 (27.42)		712 (32.40)		89 (25.57)		811 (31.87)			
	One	206 (20.77)		390 (17.76)		69 (19.83)		484 (19.02)			
	Two	427 (43.05)		868 (39.50)		162 (46.56)		993 (39.02)			
	Three	79 (7.96)		173 (7.88)		27 (7.75)		203 (7.97)			
	More than four	8 (0.80)		54 (2.46)		1 (0.29)		54 (2.12)			
Occupation	Employed	649 (65.43)		1407 (64.05)		286 (82.18)		1729 (67.94)			
	Unemployed	343 (34.57)		790 (35.95)		62 (17.82)		816 (32.06)			

SD: standard deviation. Monthly income: monetary unit is 10,000 KRW, 1 USD = 1134 KRW (Korean won, 06-21-2021). Education: educational status (for example, ≥ elementary means up to and beyond people that graduated elementary school, and none means people who did not even graduate elementary school). Marital status: others include bereaved and divorced. Occupation: occupational status (for example, unemployed means people who do not get paid, such as housewives, students, jobless, and others). Superscripts A, B, C, and D indicate statistically significant differences that exist between the groups. *** *p* < 0.001.

**Table 2 ijerph-18-10855-t002:** Characteristics for participants who were injured during PA in a year.

Variables		Family (*n* = 11) ^A^		Friend (*n* = 52) ^B^		Club Member (*n* = 37) ^C^		Alone (*n* = 31) ^D^		*F*-Value	Post Hoc
		No. (%)	Mean ± SD	No. (%)	Mean ± SD	No. (%)	Mean ± SD	No. (%)	Mean ± SD		
Gender	Male	6 (54.55)		39 (75.00)		35 (94.59)		15 (48.38)			
	Female	5 (45.45)		13 (25.00)		2 (5.41)		16 (51.62)			
Age	≤40	2 (18.18)	35.00 ± 2.83	25 (48.08)	27.80 ± 7.01	17 (45.95)	31.88 ± 5.06	12 (38.71)	25.67 ± 5.05	5.29 **	A > B, C
	41–50	1 (9.09)	50.00 ± 0.00	7 (13.46)	47.00 ± 2.94	6 (16.22)	44.67 ± 2.73	7 (22.58)	46.00 ± 2.16		
	51–70	4 (36.36)	59.50 ± 3.11	15 (28.85)	58.20 ± 5.51	14 (37.84)	56.64 ± 5.27	10 (32.26)	58.90 ± 6.03		
	≥71	4 (36.36)	72.75 ± 1.50	5 (9.62)	76.60 ± 6.23	0 (0.00)	0.00 ± 0.00	2 (6.45)	77.50 ± 6.36		
Monthly income	<290	6 (54.55)	133.33 ± 27.33	12 (23.077)	205.00 ± 60.38	2 (5.41)	245.00 ± 63.64	7 (22.58)	240.00 ± 80.21	2.96 *	C > A, B
	290–400	2 (18.18)	372.50 ± 38.89	17 (32.69)	354.71 ± 35.02	9 (24.32)	350.00 ± 38.41	7 (22.58)	361.43 ± 30.24		
	400–500	1 (9.09)	420.00 ± 0.00	10 (19.23)	466.00 ± 39.22	8 (21.62)	478.75 ± 27.48	7 (22.58)	465.71 ± 20.70		
	>500	2 (18.18)	715.00 ± 261.63	13 (25.00)	573.08 ± 48.89	18 (48.65)	593.89 ± 82.26	10 (32.26)	570.00 ± 38.87		
Education	None	1 (1.92)		0 (0.00)		0 (0.00)		0 (0.00)			
	≥Elementary	3 (5.77)		3 (27.27)		0 (0.00)		1 (3.23)			
	≥Middle	2 (3.85)		1 (9.09)		0 (0.00)		2 (6.45)			
	≥High	14 (26.92)		4 (36.36)		10 (27.00)		12 (38.71)			
	≥College	32 (61.54)		3 (27.27)		27 (73.00)		16 (54.61)			
Marital status	Married	11 (100.00)		30 (57.69)		31 (83.78)		19 (67.29)			
	Not married	0 (0.00)		18 (34.62)		6 (16.22)		10 (32.26)			
	Others	0 (0.00)		4 (7.69)		0 (0.00)		2 (6.45)			
Household size	Single	0 (0.00)		5 (9.62)		0 (0.00)		2 (6.45)			
	Two	6 (54.55)		9 (17.31)		5 (13.51)		6 (19.35)			
	Three	3 (27.27)		12 (23.08)		9 (24.32)		12 (38.71)			
	Four	1 (9.09)		25 (48.08)		20 (54.05)		9 (29.03)			
	Five	1 (9.09)		1 (1.92)		3 (8.11)		1 (3.23			
	More than six	0 (0.00)		0 (0.00)		0 (0.00)		1 (3.23)			
Number of children	No child	0 (0.00)		21 (40.38)		9 (24.32)		11 (35.48)			
	One	3 (27.27)		10 (19.23)		7 (18.92)		5 (16.13)			
	Two	4 (36.36)		17 (32.69)		17 (45.95)		15 (48.39)			
	Three	4 (36.36)		2 (3.85)		4 (10.81)		0 (0.00)			
	More than four	0 (0.00)		2 (3.85)		0 (0.00)		0 (0.00)			
Occupation	Employed	4 (36.36)		33 (63.46)		33 (89.19)		15 (48.39)			
	Unemployed	7 (63.64)		19 (36.54)		4 (10.81)		16 (51.61)			

SD: standard deviation. Monthly income: monetary unit is 10,000 KRW, 1 USD = 1134 KRW (Korean won, 06-21-2021). Education: educational status (for example, ≥ elementary means up to and beyond people that graduated elementary school, and none means people who did not even graduate elementary school). Marital status: others include bereaved and divorced. Occupation: occupational status (for example, unemployed means people who do not get paid, such as housewives, students, jobless, and others). Superscripts A, B, C, and D indicate statistically significant differences that exist between the groups. * *p* < 0.05 and ** *p* < 0.01.

**Table 3 ijerph-18-10855-t003:** Multinomial logistic regression analysis.

Variable		Injury Group (*n* = 131)					
		β	S.E	Sig.	OR	95% CI	
						Lower	Upper
Gender	Male	0.531	0.211	0.012 *	1.700	1.125	2.568
	Female				1.000	1.000	1.000
Participation mode	Family	0.006	0.355	0.986	1.006	0.501	2.016
	Friend	0.580	0.232	0.013 *	1.785	1.132	2.814
	Club member	1.654	0.272	0.000 ***	5.228	3.069	8.905
	Alone				1.000	1.000	1.000
PA intensity	Vigorous-intensity PA	1.525	0.314	0.000 ***	4.594	2.483	8.501
	Moderate-intensity PA	0.542	0.285	0.057	1.718	0.983	3.004
	Light-intensity PA				1.000	1.000	1.000
PA duration	Above 60 min	0.215	0.288	0.455	1.239	0.705	2.179
	60 min	−0.096	0.294	0.743	0.908	0.510	1.614
	Under 60 min				1.000	1.000	1.000

Reference group: not-injured group; * *p* < 0.05, *** *p* < 0.001. S.E: standard error; OR: odds ratio; CI: confidence interval; and PA: physical activity. PA duration refers to the average duration of one’s participation in PA, expressed in minutes, and groups of this were divided into quartiles.

## Data Availability

The datasets used and/or analyzed during the current study are available from the corresponding author on reasonable request.

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
