# Peer review of "When Do Individuals Get More Injured? Relationship between Physical Activity Intensity, Duration, Participation Mode, and Injury"

_ijerph, 2021, doi:10.3390/ijerph182010855_

Round 1

Reviewer 1 Report

Very interesting research.

Good review of and approach to the question, and a good use of references.

With respect to the methodology, it is not clear whether the total of 9,000 Koreans that participated in the Korean Survey of Citizens' 16 Sports Participation project were aged 18 – 89 (l. 16) or older than 10 years (l. 98). Besides, even though the size of the sample of the study  is fairly impressive as representative of the 9,000 Koreans that participated in the aforementioned study, I miss the degree of its representativity of the whole population of Koreans, either aged 18 – 89 or older than 10 years which, of course, would be much lower.

Another minor point I would like to highlight is the advisability of adding, probably in the discussion section, one or two paragraphs referring to the huge practical implications of the study in terms of cost-benefit analysis. For instance, if the authors state that regular PA of individuals has positive health consequences (l. 35) they should comment on how those benefits can be measured (less illness, longer life, well being…) and what could be the costs of injuries due to excessive PA, in terms of additional expenses for public and/or private health services.  

Besides, I miss the enunciation of possible injury prevention strategies that might help reducing injuries due to PA participation modes or PA intensity (e.g. educational programmes).

On the whole, results and conclusions seem to complement previous related findings on the matter. The methodology, data analysis and results sections all were good as well as the discussion section.

The present study undoubtedly adds insight to what is known about the subject matter and opens up new perspectives for further research.

Author Response

Responses to Editor and Reviewer Comments

We would like to thank the reviewers for the insightful comments and constructive criticism of the manuscript. We have addressed each of the specific concerns raised by the reviewers in the order that they were discussed in the comments. The requested changes have been made and designated in the red italicized text to facilitate the review. Line numbers are also provided for some responses (the line numbering restarts each page).

Reviewer Comments:
Reviewer #1:

  1. With respect to the methodology, it is not clear whether the total of 9,000 Koreans that participated in the Korean Survey of Citizens' 16 Sports Participation project were aged 18 – 89 (l. 16) or older than 10 years (l. 98).

Sorry for the confusion about this. We revised the number of years.

In line 16: “A total of 9,000 Koreans aged 10 – 89 years participated in the Korean Survey of Citizens' Sports Participation project.”

  1. Besides, even though the size of the sample of the study is fairly impressive as representative of the 9,000 Koreans that participated in the aforementioned study, I miss the degree of its representativity of the whole population of Koreans, either aged 18 – 89 or older than 10 years which, of course, would be much lower.

Thanks for mention this issue. We revised some sentences that explaining study participants.

In line 106: “This study utilized the Korean Survey of Citizens’ Sports Participation, which was conducted from September to November 2019 with 9000 Korean, older than 10 years. Stratified multistage cluster sampling using 17 regions, sex, and age groups was used in the survey to obtain a nationally representative sample from the whole Korean population (i.e., approximately 50 million).

Among the participants, this study excluded some respondents who did not respond to the question regarding PA participation modes (n = 2429), and those under 18 years old were excluded (n = 489) to focus on the association in adults. Therefore, 6082 participants’ data were utilized for the analysis. Additionally, all of the participants were categorized into four groups, < 40 (n = 2250), 41 – 50 (n = 1121), 51 – 70 (n = 2071), and > 71 (n = 640). As the survey did not collect any private information from respondents, such as participants’ names, social security numbers, or home addresses, ethical approval was not required. All study procedures were approved by the Korean Ministry of Culture, Sports, and Tourism.”

  1. Another minor point I would like to highlight is the advisability of adding, probably in the discussion section, one or two paragraphs referring to the huge practical implications of the study in terms of cost-benefit analysis. For instance, if the authors state that regular PA of individuals has positive health consequences (l. 35) they should comment on how those benefits can be measured (less illness, longer life, well-being…) and what could be the costs of injuries due to excessive PA, in terms of additional expenses for public and/or private health services.  

Thanks for catching these issues. We added a sentence in a paragraph as suggested.

In line 328: “With the efforts of health-related institutions in many countries, it became well known that participating PA would help individuals to improve their health and obtain additional health benefits (i.e., longer life, less illness, and well-being). Practically, one of the studies regarding the association between PA and medical cost indicated that a $1 investment for participating PA resulted in $2.94 in direct medical cost-benefit [49]. Because of the positive benefits of PA, many individuals are more likely to engage in PA vigorously and frequently to obtain additional benefits. However, as previously mentioned, the increased volumes of PA are likely to increase the possibility of getting an injury. A study concerning the association between PA and the medical cost for injury treatments showed that the average cost of the treatment for sports injury was $1,510 per injury, which could offset the benefits of PA [50]. Based on these studies, individuals should take into account the potential results derived from PA either positively and negatively so that individuals can take more additional benefits from PA. By using this present study’s results, individuals can consider how they minimize the medical cost of injury and how they can maximize the benefits from PA.”

  1. Besides, I miss the enunciation of possible injury prevention strategies that might help reducing injuries due to PA participation modes or PA intensity (e.g. educational programmes).

Thank you for catching this! We added sentences explaining strategies for preventing injury.

In line 280: Also, health practitioners, instructors, and coaches of sports clubs should educate participants on how they can prevent injuries through educational programs prior to PA. Furthermore, the association between PA participation mode and injury occurrence requires further studies on whether other PA participation modes that were not included in the present study would potentially prevent or cause injuries.

In line 306: “Additionally, if health practitioners, instructors, and coaches exist, they should adjust the intensity and duration of PA depending on the purpose of participating in PA and individuals’ fitness status.”

In line 323: “Therefore, in these circumstances, more specific recommendations and policies (e.g. educational and promotional programs) [48] should be implemented in sports clubs to prevent injuries especially for males. Also, It is necessary to develop the PA promotional events (e.g. opening female’s sports clubs and events) to increase females’ participation in sports clubs.”

Reviewer 2 Report

In this paper, the authors aimed to explore the association between injury rate and physical activity participation modes (family, friends, club members and alone), as well as the relationship between injury and physical activity volume (intensity and duration). The multinomial logistic regression was utilized to acquire the associations between the two above-mentioned variable pairs. A large number of participants was obtained from the Korean Survey of Citizen’s Sports Participation project. Some interesting findings were presented. However, there were some issues or problems that the authors might need to modify to improve the quality of the paper.

  1. The main problem of this manuscript was that the data was not well organized and presented. The authors firstly defined four participation modes but further classified each mode into several subgroups based on different factors, such as gender, age, monthly income, education et. al. This further classification was not necessary because it did not contribute to investigating the association between injury and PA mode or between injury and PA volume and might confuse the readers. The authors just described the statistical results of different subgroups but did not discuss the connections of these results with injury. Meanwhile, it made the data really fragmented that the readers might spend too much time to get understand the data. For example, in table 2 the number of male and female participants of the four PA modes who injured in the last year were displayed. However, in the context (Line 186), the authors did not use these numbers, but compared the total male and female numbers (95 and 36) which were not presented in the table.

  1. The authors emphasized in the abstract and introduction that the purpose of this study was to investigate the relationship between injury and PA modes as well as PA volume. However, in table 3 and the second paragraph in page 8, the gender factor was also analyzed and described. Therefore, the authors should modify the corresponding parts in abstract and introduction to add the gender.

  1. In the discussion part (Line 279-281), the authors proposed a hypothesis that the high injury rates in the sports club member and friend modes were due to overburden in muscles, bones, and nerves. To validate this hypothesis, the authors should analyze the PA volumes of the four pre-defined PA modes, especially the PA intensity which was confirmed to be related to injury rate in the same study.

  1. The authors should add a paragraph to introduce the multinomial logistic regression method, especially the basic rationale and the meaning of the outcome parameters.

  1. The discussion part was not well prepared. The authors just repeated the results and provide some similar studies but did not compare the results with previous studies and did not provide sufficient explanations. This should be improved.

  1. There were many typos and grammatical mistakes.

Typos:

Line 16: A total of 9000 Korean aged 18-89 years… However, in Line 18, the authors wrote “those under 18 years old were excluded…”

Line 19: Analysis of variance was also used to compare… The word also should be removed.

Line 45: is the risk of musculoskeletal injury… The word risk should be removed.

Line 111: the options of the question were… The word question should be answer.

Grammatical mistakes:

Sentence in Line 58: The major participation modes…

Sentence in Line 78: Similarly, Malisoux’s study demonstrated that the week of …

Sentence in Line 85: to examine the important association whether the volumes…

Sentence in Line 249: The purpose of this study was to examine the differences…

Author Response

Responses to Editor and Reviewer Comments

We would like to thank the reviewers for the insightful comments and constructive criticism of the manuscript. We have addressed each of the specific concerns raised by the reviewers in the order that they were discussed in the comments. The requested changes have been made and designated in the red italicized text to facilitate the review. Line numbers are also provided for some responses (the line numbering restarts each page).

Reviewer Comments:
Reviewer #2:

  1. The main problem of this manuscript was that the data was not well organized and presented. The authors firstly defined four participation modes but further classified each mode into several subgroups based on different factors, such as gender, age, monthly income, education et. al. This further classification was not necessary because it did not contribute to investigating the association between injury and PA mode or between injury and PA volume and might confuse the readers. The authors just described the statistical results of different subgroups but did not discuss the connections of these results with injury.

Thanks for the comment. We understand what you pointed out, but the additional classification (i.e., gender, age, monthly income, and education et al.), which did not contribute to the main results of this study, was to compare the participation modes’ socioeconomic status. Although we did not discover the direct association between those and injury, differences of numbers in some factors were found. In short, these results may help readers understand the characteristics of participation modes.

Additionally, we discussed the gender difference, which resulted from Table 1 and 2, in the discussion section.

  1. Meanwhile, it made the data really fragmented that the readers might spend too much time to get understand the data. For example, in table 2 the number of male and female participants of the four PA modes who injured in the last year were displayed. However, in the context (Line 186), the authors did not use these numbers, but compared the total male and female numbers (95 and 36) which were not presented in the table.

Thanks for catching these issues. We changed the sentence.

In line 200: “the number of male participants in the friend and club member group was three and 17 times higher than females, respectively.”

  1. The authors emphasized in the abstract and introduction that the purpose of this study was to investigate the relationship between injury and PA modes as well as PA volume. However, in table 3 and the second paragraph in page 8, the gender factor was also analyzed and described. Therefore, the authors should modify the corresponding parts in abstract and introduction to add the gender.

Thank you for your comment! We revised the sentence in the abstract and added a paragraph in the introduction.

In line. 74: “Several studies indicated that gender is also a critical factor when individuals participated in PA. One of the studies regarding participants' motivation and context preferences of PA showed significant differences in gender such as preferred environments, characteristics of PA, and reasons for participating in PA [23]. In addition, Christopher et al. [2] explained that female’s anxiety sensitivity might affect their volume of PA. In the same context, many studies indicated that males are more likely to have injuries during PA than females because males tend to participate in strenuous exercise [3, 4]. Therefore, the extent to which interventions for the injury occurrence and reduction of anxiety sensitivity should be taken into account for the gender gap in PA.

  1. In the discussion part (Line 279-281), the authors proposed a hypothesis that the high injury rates in the sports club member and friend modes were due to overburden in muscles, bones, and nerves. To validate this hypothesis, the authors should analyze the PA volumes of the four pre-defined PA modes, especially the PA intensity which was confirmed to be related to injury rate in the same study.

Thanks for catching this! We revised the sentence.

In line. 275: “Based on the result of Table 3 and Figure B, we speculated that participating in PA in a sports club and/or with friends causes increasing the volumes of PA, especially intensity. It might lead to overburdening in muscles, bones, and nerves through frequent contact with others via strenuous movements.”

  1. The authors should add a paragraph to introduce the multinomial logistic regression method, especially the basic rationale and the meaning of the outcome parameters.

Thank you for your comment! We revised a part of an explanation of the multinomial logistic regression.

In Line. 173: “In addition, multinomial logistic regression was utilized to examine the association between injury occurrence and gender, PA participation modes, and PA volumes (i.e., intensity and duration), and compared injury occurrence in each variable's subgroup.”

  1. The discussion part was not well prepared. The authors just repeated the results and provide some similar studies but did not compare the results with previous studies and did not provide sufficient explanations. This should be improved.

Thank you for mention it! We addressed simple direct explaination by comparing the results with previous studies but realized that the statement in the discussion section needed to add sufficient explanations. Therefore, we revised some of the statements to help readers to understand the concepts.

In line 280: “Also, health practitioners, instructors, and coaches of sports clubs should educate participants on how they can prevent injuries through educational programs prior to PA. Furthermore, the association between PA participation mode and injury occurrence requires further studies on whether other PA participation modes that were not included in the present study would potentially prevent or cause injuries.

In line. 291: “Although a few studies are insisting that PA duration is related to the increased rate of injury, those studies are differentiated from the present study because those studies’ participants were in particular environments (i.e., military, athletes, and specific places).”

In line 306: “Additionally, if health practitioners, instructors, and coaches exist, they should adjust the intensity and duration of PA depending on the purpose of participating in PA and individuals’ fitness status.”

In line 323: “Therefore, in these circumstances, more specific recommendations and policies (e.g. educational and promotional programs) [48] should be implemented in sports clubs to prevent injuries especially for males. Also, It is necessary to develop the PA promotional events (e.g. opening female’s sports clubs and events) to increase females’ participation in sports clubs.”

In line 328: “With the efforts of health-related institutions in many countries, it became well known that participating PA would help individuals to improve their health and obtain additional health benefits (i.e., longer life, less illness, and well-being). Practically, one of the studies regarding the association between PA and medical cost indicated that a $1 investment for participating PA resulted in $2.94 in direct medical cost-benefit [49]. Because of the positive benefits of PA, many individuals are more likely to engage in PA vigorously and frequently to obtain additional benefits. However, as previously mentioned, the increased volumes of PA are likely to increase the possibility of getting an injury. A study concerning the association between PA and the medical cost for injury treatments showed that the average cost of the treatment for sports injury was $1,510 per injury, which could offset the benefits of PA [50]. Based on these studies, individuals should take into account the potential results derived from PA either positively and negatively so that individuals can take more additional benefits from PA. By using this present study’s results, individuals can consider how they minimize the medical cost of injury and how they can maximize the benefits from PA.”

  1. There were many typos and grammatical mistakes.

We fixed all typos and grammatical errors via a native speaker.

Typos:

Line 16: A total of 9000 Korean aged 18-89 years… However, in Line 18, the authors wrote “those under 18 years old were excluded…”

Thank you for mention it! We revised the number from 18 to 10

Line 19: Analysis of variance was also used to compare… The word also should be removed.

 Thanks for the suggestion! We erased the word.

Line 45: is the risk of musculoskeletal injury… The word risk should be removed.

 Thanks for catching it! We erased the words “the risk of”

Line 111: the options of the question were… The word question should be answer.

Thanks for your advice! We changed the word.

Grammatical mistakes:

Sentence in Line 58: The major participation modes…

 Thanks for mention the issue! We revised the words.

In line. 58: “During PA, participation modes (i.e., family, friends, club members, and alone) are important factors to influence PA volumes (i.e., intensity and duration), and the PA volumes are directly related to inducing injuries.”

Sentence in Line 78: Similarly, Malisoux’s study demonstrated that the week of …

Thanks for catching it! We revised the sentence.

In line. 88: “Similarly, Malisoux’s study demonstrated that intensity during the week before an injury was significantly higher than the 4 preceding weeks.”

Sentence in Line 85: to examine the important association whether the volumes…

 Thank you for your notice! We changed some words in the sentence.

In line. 94: “To examine the important association of whether the volumes (i.e., intensity and duration) and modes of physical activity affect injury occurrence.”

Sentence in Line 249: The purpose of this study was to examine the differences…

Thanks for your advice! We checked the mistake and revised it.

In line. 245: “The purpose of this study was to examine the differences in injury rates in each PA participation mode and the critical association of whether the volumes (i.e., intensity and duration) and modes of PA affect injury occurrence.”

Additionally, we revised some other parts in this paper for better information.

In line 106: “This study utilized the Korean Survey of Citizens’ Sports Participation, which was conducted from September to November 2019 with 9000 Korean, older than 10 years. Stratified multistage cluster sampling using 17 regions, sex, and age groups was used in the survey to obtain a nationally representative sample from the whole Korean population (i.e.,  approximately 50 million).

Among the participants, this study excluded some respondents who did not respond to the question regarding PA participation modes (n = 2429), and those under 18 years old were excluded (n = 489) to focus on the association in adults. Therefore, 6082 participants’ data were utilized for the analysis. Additionally, all of the participants were categorized into four groups, < 40 (n = 2250), 41 – 50 (n = 1121), 51 – 70 (n = 2071), and > 71 (n = 640). As the survey did not collect any private information from respondents, such as participants’ names, social security numbers, or home addresses, ethical approval was not required. All study procedures were approved by the Korean Ministry of Culture, Sports, and Tourism.”

Round 2

Reviewer 2 Report

Referring to my previous review comments, the authors presented point-to-point responses. I really appreciate the efforts of the authors made to revise the manuscript. After rereading the manuscript, I feel that the quality of the paper has been improved impressively. However, the major concern about the data presentation still exists.

  1. Overall, the ANOVA results were presented incompletely and vaguely. The authors did not display the mean and SD values of the group characteristics (in table or figure formats), which were essential for readers to understand the comparison results. In the context, the authors only provided the incomplete results of gender (Line 182) and monthly income (Line 187) but did not mention the comparison results (significantly different or not) of other factors (age, education, marital status, et al).

  1. Meanwhile, the expression about the ANOVA test was also unclear. The authors did not mention whether post-hoc test was used to reveal the difference between PA modes if ANOVA test showed significant difference. In general, ANOVA just told us whether there was significant difference among the four PA modes, but we did not know which two groups showed significant difference. Hence, we need post-hoc test to find the group pairs that has statistically significant difference (for example the age of family group was significantly higher than club member group but had no significant difference with the rest two groups).

  1. Line 182: “Overall, the number of males and females showed non-significant differences in family, friend and alone groups (p>0.05) …” The authors should clarify what kind of statistical method was used to compare the number of males and females between different PA mode groups because the ANOVA method could not be used to compare just two numbers.

  1. Line 227: “According to Figure 1 (A), when males participated in PA with friends, there was the highest injury occurrence (41.1%) and followed by exercising with club members (36.8%), alone (15.8%), and family group (6.3%).” In this sentence and the following sentence, the authors used the percentage of injuries in each PA mode among the whole injury number of all the four participation modes as the injury occurrence. However, this calculation method ignored the basis number of each PA mode (for example, males: 430 for family, 1153 for friend, 275 for club member and 1251 for alone). Therefore, this percentage data obtained by absolute injury number could not reflect the real injury risk of each PA mode. The authors should use the injury rate (injury number of male or female divided by the corresponding total male or female participant number) to present the risk of each PA mode (for example, males: 1.4% for family, 3.4% for friend, 12.7% for club member, and 1.2% for alone). The following sentence has the same problem.

  1. Line 238: “By using Bonferroni post-hoc analysis, significant differences were found between the participation modes…” The authors should present the results in detail with all the significance levels and mean values of all the 6 comparison pairs (family-friend, family-club member, family-alone, friend-club member, friend-alone, and club member-alone). Additionally, Bonferroni is a correction method to avoid high chance of type I error in statistical analysis rather than a post-hoc test. Therefore, the authors should clarify the post-hoc test method that they used to do the multiple pair comparisons.

  1. Line 245: “The purpose of this study was to examine the differences in injury rates in each PA participation mode…” However, the authors did not provide the injury rates of the four PA participation modes. It would be better if the authors calculate the overall injury rates of each mode as well as the injury rates of females and males.

  1. The data provided in this study was not well used and discussed in the discussion part. For example, when talking about the higher risk of male participants than females (Line 248), the authors just referred to other studies which attributed the higher injury rates to the tendency of males to participate in more competitive sports. However, the study contained the data of PA volumes (intensity and duration) of both male and female subjects in the four modes. Therefore, the authors should also try to analyze whether the higher injury occurrence of males were related to the PA intensity or duration. In this way, the data of this study could better support the conclusions the authors proposed rather than just use references.

  1. Minor revisions:

  • The format of Table 2 should be modified to match with Table 1 (which is more compact).

  • The “PA volumes of participants” in Figure 1 should be changed to “PA intensities of participants”. Meanwhile, the sentence in Line 232 “Figure 1 (B) showed the comparison with the PA volumes (i.e., intensity and duration) by…” should also be modified.

  • The order of the four PA modes in Figure 1 should be corrected to be from left to right: family, friend, club member and alone to match with the order in the tables.

  • Line 200: “The number of male participants in the friend…” The font of words “participants” and “respectively” were different from other parts.

Author Response

Responses to Editor and Reviewer Comments

We would like to thank the reviewers for the insightful comments and constructive criticism of the manuscript. We have addressed each of the specific concerns raised by the reviewers in the order that they were discussed in the comments. The requested changes have been made and designated in the red italicized text to facilitate the review. Line numbers are also provided for some responses (the line numbering restarts each page).

Reviewer Comments (Round 2):
Reviewer #2:

  1. Overall, the ANOVA results were presented incompletely and vaguely. The authors did not display the mean and SD values of the group characteristics (in table or figure formats), which were essential for readers to understand the comparison results. In the context, the authors only provided the incomplete results of gender (Line 182) and monthly income (Line 187) but did not mention the comparison results (significantly different or not) of other factors (age, education, marital status, et al).

Thank you for your comment! We understood what you mentioned. We displayed the mean and SD in Table 1 and 2, especially for age and monthly income variables. The mean and SD of the other variables (i.e., education, marital status, et al) were incalculable because those were the nominal variables.

Meanwhile, the expression about the ANOVA test was also unclear. The authors did not mention whether post-hoc test was used to reveal the difference between PA modes if ANOVA test showed significant difference. In general, ANOVA just told us whether there was significant difference among the four PA modes, but we did not know which two groups showed significant difference. Hence, we need post-hoc test to find the group pairs that has statistically significant difference (for example the age of family group was significantly higher than club member group but had no significant difference with the rest two groups).

Thank you for your advice! We wrote more explanations about results and added F-value and Post-hoc results in Table 1 and 2 by using superscripts A, B, C, and D for the group significant difference indicators.

In line 203: “The number of participants under 50 years old was relatively higher in all groups than that over 51 years old group, but there were no significant differences in each participation mode (p > 0.05). In the monthly income, more than half of each group earned more than 2900 thousand KRW (USD 2569), and the club member group revealed significant differences between each of the other groups (i.e., family, friend, and alone) (p < 0.001). There were no significant differences between the three groups (i.e., family, friend, and alone) (p > 0.05).”

In line 218: “In participants’ age, an age difference was found between the family group and two groups (club member and friend group). The result indicated that the age of the family group (mean = 59.00 years) was significantly higher than the club member (mean = 43.32 years) (p < 0.05) and the friend group (mean = 43.85 years) (p < 0.05), but there were no significant differences between the other pairs (p > 0.05). All group’s monthly incomes also presented differences in Table 2. Half of the monthly income of the family and friend group was under 4 million KRW, and the club member and alone group of that was above 4 million KRW. The mean difference (unit: 10,000 KRW) between the club member and two groups (friend and family) was 94.66 (p < 0.05) and 182.17 (p < 0.01), respectively, and there were no significant differences between the other pairs.”

  1. Line 182: “Overall, the number of males and females showed non-significant differences in family, friend and alone groups (p > 0.05) …” The authors should clarify what kind of statistical method was used to compare the number of males and females between different PA mode groups because the ANOVA method could not be used to compare just two numbers.

Thank you for your comment! We intended to compare the sheer number of females and males and used an independent t-test to examine the gender difference in continuous variables (i.e., age, and income level) but other variables were categorical variables so we also needed to performed chi-square tests to examine the proportion difference. But the information is not critical for the main purpose of this study so we decided to remove the statistical information for the reader to simply understand the characteristics of participants. We explained it in the text instead.

  1. Line 227: “According to Figure 1 (A), when males participated in PA with friends, there was the highest injury occurrence (41.1%) and followed by exercising with club members (36.8%), alone (15.8%), and family group (6.3%).” In this sentence and the following sentence, the authors used the percentage of injuries in each PA mode among the whole injury number of all the four participation modes as the injury occurrence. However, this calculation method ignored the basis number of each PA mode (for example, males: 430 for family, 1153 for friend, 275 for club member and 1251 for alone). Therefore, this percentage data obtained by absolute injury number could not reflect the real injury risk of each PA mode. The authors should use the injury rate (injury number of male or female divided by the corresponding total male or female participant number) to present the risk of each PA mode (for example, males: 1.4% for family, 3.4% for friend, 12.7% for club member, and 1.2% for alone). The following sentence has the same problem.

Thank you for your comment! We changed the percentage into the number of injury occurrences and added sentences explaining the percentage.

In line 255: “Specifically, Figure 1 revealed the comparison of the number of injury occurrences (Figure 1, A) and PA volumes (Figure 1, B) for each PA participation mode. According to Figure 1 (A), when males participated in PA with friends, there was the highest injury occurrence (n = 39) and followed by club members (n = 35), alone (n = 15), and family group (n = 6). Meanwhile, the percentage of injury occurrence was highest in the club member group (12.7%) followed by the friend (3.4%), family (1.4%), and alone group (1.2%). In female, the club member group had the lowest cases (n = 2) following the family (n = 5), friend (n = 13), and alone group (n = 16). But, in percentage context, the club member group (2.7%) had the highest percentage of injury occurrence followed by the friend (1.2%), alone (1.2%), and family group (0.9%).”

  1. Line 238: “By using Bonferroni post-hoc analysis, significant differences were found between the participation modes…” The authors should present the results in detail with all the significance levels and mean values of all the 6 comparison pairs (family-friend, family-club member, family-alone, friend-club member, friend-alone, and club member-alone). Additionally, Bonferroni is a correction method to avoid high chance of type I error in statistical analysis rather than a post-hoc test. Therefore, the authors should clarify the post-hoc test method that they used to do the multiple pair comparisons.

Thank you for the mention! We revised the sentence for detail information.

In line 271: “By using Bonferroni post-hoc analysis, significant mean differences (p < 0.01) were found between the club member and friend (males: 384.39, females: 527.74 MET-min/week), club member and family (males: 484.30, females: 769.18 MET-min/week), club member and alone (males: 209.58, females: 526.74)), family and alone group (males: 274.71, females: 242.43 MET-min/week), friend and alone (males: 174.81 MET-min/week), and friend and family group (females: 241.43 MET-min/week). Between the friend and family group (mean difference = 99.90) in males and between the friend and alone group (mean difference = 0.99) in females did not show significant differences.”

  1. Line 245: “The purpose of this study was to examine the differences in injury rates in each PA participation mode…” However, the authors did not provide the injury rates of the four PA participation modes. It would be better if the authors calculate the overall injury rates of each mode as well as the injury rates of females and males.

Thank you for your comment! We understand what you mentioned. We think injury rates in each participation mode can be found in Figure A that we recently added some percentage explanations.

In line 259: “Meanwhile, the percentage of injury occurrence was highest in the club member group (12.7%) followed by the friend (3.4%), family (1.4%), and alone group (1.2%). In female, the club member group had the lowest number (n = 2) following the family (n = 5), friend (n = 13), and alone group (n = 16). But, in percentage context, the club member group (2.7%) had the highest percentage of injury occurrence followed by the friend and alone (1.2%), and family group (0.9%).”

  1. The data provided in this study was not well used and discussed in the discussion part. For example, when talking about the higher risk of male participants than females (Line 248), the authors just referred to other studies which attributed the higher injury rates to the tendency of males to participate in more competitive sports. However, the study contained the data of PA volumes (intensity and duration) of both male and female subjects in the four modes. Therefore, the authors should also try to analyze whether the higher injury occurrence of males were related to the PA intensity or duration. In this way, the data of this study could better support the conclusions the authors proposed rather than just use references.

Thanks for your advice! We revised the second paragraph in the discussion part.

In line 289: “Not only the duration of PA but also males’ PA intensity is much higher than that of females [32, 33]. According to a study regarding injuries of runners in perspective of gender, males were at higher risk than females suffering from running-related injuries, particularly; 1) in the case of having running experience of 0 – 2 years, 2) a history of previous injuries, restarting running, and 3) excessive running distance of more than 40 miles per week [25]. These results might be from males being more active than females and they tend to participate in more competitive sports [34]. Additionally, higher PA intensity and duration of males might be potential risk factors of increased injury risk rate.

  1. Minor revisions:

  • The format of Table 2 should be modified to match with Table 1 (which is more compact).

Thanks for the mention! We revised the Table format!

  • The “PA volumes of participants” in Figure 1 should be changed to “PA intensities of participants”. Meanwhile, the sentence in Line 232 “Figure 1 (B) showed the comparison with the PA volumes (i.e., intensity and duration) by…” should also be modified.

Thank you for your suggestion! We calculated METs by PA intensity and duration, so we think it is better to write as PA volumes rather than PA intensity.

  • The order of the four PA modes in Figure 1 should be corrected to be from left to right: family, friend, club member and alone to match with the order in the tables.

Thanks for issuing this! We revised the order of the four PA modes in Figure 1.

  • Line 200: “The number of male participants in the friend…” The font of words “participants” and “respectively” were different from other parts.

Thank you for the finding! We revised the font.